# Capsaicin Modulates Hepatic and Intestinal Inflammation and Oxidative Stress by Regulating the Colon Microbiota

**DOI:** 10.3390/antiox13080942

**Published:** 2024-08-02

**Authors:** Xiaotong Pang, Xin Wei, Yanyan Wu, Shanshan Nan, Jiaqi Feng, Fang Wang, Min Yao, Cunxi Nie

**Affiliations:** 1College of Animal Science and Technology, Shihezi University, Shihezi 832000, China; pxt1524896983@sina.com (X.P.); weix20220222@163.com (X.W.); 1wuyan1@163.com (Y.W.); 18016862099@163.com (S.N.); 20212013017@stu.shzu.edu.cn (J.F.); 20222013004@stu.shzu.edu.cn (F.W.); 2School of Medicine, Shihezi University, Shihezi 832000, China

**Keywords:** liver, gut, oxidative stress, inflammation, intestinal microbiota, capsaicin

## Abstract

We aimed to investigate the role of capsaicin (CAP) in modulating lipopolysaccharide (LPS)-induced hepatic and intestinal inflammation, oxidative stress, and its colonic microflora in mice. Thirty healthy male Kunming mice with similar body weights were randomly assigned to three groups: the control group (CON), the LPS group, and the CAP group, with ten mice in each group. The CON and the LPS groups received a daily dose of normal saline, respectively, while the CAP group received an equivalent dose of CAP. On the 28th day of the experiment, the LPS and the CAP groups were intraperitoneally injected with LPS, while the CON group was injected with an equal volume of normal saline. The results lead to the following conclusions. Compared to the LPS group, CAP improved the loss of hepatic lobular structure and significantly increased the duodenal villus length and ratio of villus length to crypt depth. CAP increased hepatic and colon interleukin-10 (IL-10) and decreased IL-6, IL-1β, and tumor necrosis factor (TNF-α) levels. CAP also increased hepatic catalase (CAT), glutathione peroxidase (GSH-Px), and superoxide dismutase (SOD) expression, and decreased malondialdehyde (MDA) levels. CAP significantly increased the relative abundances of *Mucispirillum*, *Helicobacter*, *Prevotellaceae-UCG-001*, *Colidextribacter*, *unclassified-f-Oscillospiraceae*, and *Odoribacter*, some of which were closely related to hepatic and colonic immune and oxidative markers. CAP also decreased the overall content of short-chain fatty acids, except for propionic acid. Overall, CAP can regulate the colon microbiota and exert anti-inflammatory and antioxidant effects. Whether CAP exerts its anti-inflammatory and antioxidant effects by modulating the colonic microflora, mainly *Mucispirillum* spp. and *Helicobacter* spp., requires further investigation.

## 1. Introduction

The incidence of liver and intestinal metabolic diseases is steadily increasing worldwide. Both the liver and intestines, as detoxification and immune organs, respectively, are crucial in regulating human health. A significant contribution to the homeostasis of the liver and intestinal microbiota is made by the enterohepatic axis between the gastrointestinal tract and the liver. Liver disease may disrupt intestinal homeostasis. Intestinal microbiota dysbiosis and altered permeability lead to a variety of immune, inflammatory, and neoplastic intestinal and liver diseases [1], including hepatic steatosis, inflammation, fibrosis, viral hepatitis, and acute liver failure [2,3,4], as well as bowel diseases such as colorectal cancer, ulcerative colitis, and Crohn’s disease [5,6]. Thus, liver diseases and intestinal homeostatic disorders are interrelated and mutually reinforced.

Oxidative stress is the consequence of an unbalance between the generation of reactive oxygen species (ROS) and the antioxidant system. This imbalance ultimately damages cellular biomolecules (DNA, proteins, and lipids) [3]. A pathophysiological process, oxidative stress is closely associated with inflammation [7]. Lipopolysaccharide (LPS) acts as a hepatic Toll-like receptor 4 sensor that promotes oxidative stress [8]. ROS generated by oxidative stress can mediate the expression of pro-inflammatory genes and initiate chronic inflammation via intracellular signaling pathways [7]. Increased inflammation can in turn trigger disturbances in the intestinal microbiota. Likewise, LPS disrupts bacterial microbiota homeostasis, which in turn induces systemic inflammation through the intestinal barrier and blood circulation, promoting the release of inflammatory factors that lead to increased oxidative stress at the site of inflammation, thus creating a vicious cycle [9]. Oxidative stress, inflammation-mediated liver and intestinal diseases, and intestinal microbial dysbiosis pose challenges.

Capsaicin (CAP) is an alkaloid with antioxidant, anti-inflammatory, antimicrobial, anti-obesity, anti-cancer, analgesic, and gut-stimulating properties [10,11]. Acting as an anti-inflammatory compound, CAP enhances gastrointestinal mucosal antioxidant enzyme activity and protects the gastrointestinal tract from systemic inflammation and sepsis [12,13]. In the presence of mild systemic inflammation, acute CAP administration may alter the oxidative state of certain tissues and exert anti-inflammatory effects, preventing LPS-induced tissue damage in the liver and lungs [14]. Dietary CAP has been reported to adjust the composition and abundance of the intestinal microbiota and to influence the physiological status of mice by altering the composure of the gut microbiota [15,16]. The research that has been conducted on CAP suggests it may have an influential role in biologically active compounds and in chili peppers. Based on the anti-inflammatory and antioxidant effects exerted by CAP, the aim of this study was to explore whether CAP could attenuate LPS-induced hepatic and intestinal inflammation and alleviate hepatic oxidative stress by modulating the colon microbiota.

## 2. Materials and Methods

### 2.1. Animal Ethics Statement

This study was approved by the Biological Ethics Committee of Shihezi University (A2023-123). All experiments were performed in accordance with the relevant guidelines and regulations of the ARRIVE guidelines (https://arriveguidelines.org, accessed on 25 March 2023).

### 2.2. Chemicals and Reagents

CAP and LPS were purchased from Solarbio Co., Ltd. (Beijing, China; CAP: catalog SC8100, lot number 111K021; LPS: catalog L8880, lot number 1031Q0312). Interleukin-6 (IL-6), IL-10, IL-1β, and tumor necrosis factor (TNF-α) kits were purchased from Jiangsu Jingmei Biological Technology Co., Ltd. (Yancheng, China). Superoxide dismutase (SOD), malondialdehyde (MDA), glutathione peroxidase (GSH-Px), and catalase (CAT) kits were purchased from Suzhou Gerace Biotech Co., Ltd. (Suzhou, China). The TransZol Up Plus RNA Kit was purchased from Beijing All Style Gold Biotechnology Co., Ltd. (Beijing, China). The HiFiScript cDNA Synthesis Kit was purchased from Beijing Kangwei Century Biotechnology Co., Ltd. (Beijing, China). The PerfectStart Green qPCR SuperMix kit was purchased from Beijing AllStyle Gold Biotechnology Co., Ltd. (Beijing, China).

### 2.3. Animals and Experimental Design

As shown in Figure 1, 30 healthy male Kunming mice of similar weight were randomized to control (CON), LPS, and CAP groups (*n* = 10, 20–22 g). The CON and LPS groups were administered daily with equal doses of saline via oral gavage (0.3 mL/per mouse), while the CAP group was administered CAP (7.5 mg/kg). On the 28th day of the experiment, LPS (10 mg/kg) was injected intraperitoneally in the LPS and CAP groups, and the CON group had the same amount of saline injection. After 4 h, mice were sacrificed, and samples were collected (including liver, kidney, spleen, duodenum, jejunum, ileum, colon tissue, and colonic contents).

### 2.4. Sample Collection

Mouse body weight was noted weekly. At the end of the experiment, neck-breaking was performed on the mice, and orbital blood collection was performed. The serum of the mice was centrifuged at 3000 rpm for 10 min at 4 °C. The collected liver, kidneys, spleen, and thymus were extracted, weighed, and used to calculate organ indices. Tissues collected from the liver and small and large intestine sections were fixed in 4% paraformaldehyde, and a part of the tissue was simultaneously flash-frozen with liquid nitrogen and then transferred until further analyses.

### 2.5. Organ Indices of the Liver, Kidney, Spleen, and Thymus

The liver, kidney, spleen, and thymus were weighed, and the index of each organ was calculated by dividing the absolute weight (mg) by the body weight (g) and then multiplying by 100%.

### 2.6. Morphological Analysis of the Liver and Intestinal Tissues

The liver, duodenum, jejunum, ileum, and colon were fixed in 4% paraformaldehyde, dehydrated using an ethanol series, made transparent with xylene, and embedded in paraffin wax. Hematoxylin and eosin staining was performed on the embedded wax blocks. Liver and intestinal morphology was viewed using a microscope (Olympus BX53, Tokyo, Japan), and villus length and crypt depth were measured using the ImageJ software (version ImageJ 1.52v).

### 2.7. Liver and Colon Inflammatory Biomarker Assays

Liver and colon interleukin-6 (IL-6), IL-10, IL-1β, and tumor necrosis factor (TNF-α) levels were determined using enzyme-linked immunosorbent assay (ELISA) kits (Jiangsu Jingmei Biological Technology Co., Ltd., Yancheng, China).

### 2.8. Expression of Hepatic Oxidation/Reduction Markers

Superoxide dismutase (SOD), malondialdehyde (MDA), glutathione peroxidase (GSH-Px), and catalase (CAT) were measured as oxidation/reductions factors in the liver using a micro-enzymatic kit (Suzhou Gerace Biotech Co., Ltd., Suzhou, China) and a full-wavelength enzyme labeler (Thermo Fisher Scientific, Ltd., Shanghai, China).

### 2.9. Quantitative Real-Time Reverse-Transcription Polymerase Chain Reaction

Total RNA was obtained from liver and colon tissues by using the TransZol Up Plus RNA Kit (Beijing All Style Gold Biotechnology Co., Ltd., Beijing, China), and a Nanodrop 2000 visible spectrophotometer (Thermo Fisher) was used to assess the purity and integrity of the RNA. The HiFiScript cDNA Synthesis Kit (Beijing Kangwei Century Biotechnology Co., Ltd., Beijing, China) was used to reverse transcribe RNA into cDNA. Primers were designed using Primer 5 (Table 1) and synthesized by Youkang Biotechnology Co., Ltd. (Xinjiang, China). Using β-actin as an internal reference gene, qRT-PCR was analyzed by use of the PerfectStart Green qPCR SuperMix kit (Beijing AllStyle Gold Biotechnology Co., Ltd., Beijing, China) and a LightCycler 96 System (Roche Applied Science, Penzberg, Germany). The relative quantification of the above genes was conducted using the 2^−ΔΔCt^ test method. Amplification conditions included initial denaturation at 94 °C for 30 s, followed by 45 cycles of denaturation at 94 °C for 5 s, annealing at 60 °C for 15 s, and extension at 72 °C for 10 s.

### 2.10. Colonic 16s rRNA High-Throughput Sequencing

Total microbial genomic DNA was extracted from colon contents using the E.Z.N.A.^®^ soil DNA Kit (Omega Bio-tek, Norcross, GA, USA), followed by PCR amplification. Library building was performed using NEXTFLEX^®^ Rapid DNA-Seq Kit, and high-throughput sequencing was performed using Illumina NovaSeq PE250 (Shanghai Meiji Biomedical Science and Technology Co., Ltd., Shanghai, China). The metagenomic function was predicted using PICRUSt2 (version 2.2.0) and analyzed on the Majorbio Cloud platform. The main methods included Chao 1, Shannon, and Simpson’s indices.

### 2.11. Detection of Colonic Short-Chain Fatty Acid (SCFA) Content

The content of short-chain fatty acids (SCFAs) in the colon was analyzed by gas chromatography-mass spectrometry (Agilent Technologies 7890 B GC system, Shanghai, China). Briefly, 1500 μL of ultrapure water was added to 0.3 g of colon contents to obtain a 20% m/v slurry. The homogenate was then centrifuged at 25 °C (5000 rpm for 4 min), and the supernatant was recovered. Subsequently, to each supernatant, 100 μL of a 25% solution of metaphosphoric acid was added and vortexed for 30 s, and centrifuged at 25 °C (15,000 rpm, 15 min). The resulting supernatant was collected. It was passed through a 0.45 μm aqueous membrane. The resulting supernatant was collected and passed through a 0.45 μm aqueous membrane. The SCFA content was determined using a DB-WAX column (60 m × 250 μm × 0.25 μm) and an FID detector. The column temperature was heated from 100 °C to 180 °C at a ramp rate of 8 °C/min, held for 1 min, followed by heating from 180 °C to 200 °C at a ramping rate of 20 °C/min and then held for 5 min. Standard curves were constructed, and SCFA content was calculated.

### 2.12. Statistical Analyses

SPSS software, version 22.0, was used to analyze the data. Data are expressed as mean ± standard error (SEM), and *p* < 0.05 indicates a significant difference. Significance was determined using a one-way analysis of variance (ANOVA) with Duncan’s test. Graphs were generated using Origin 2021.

## 3. Results

### 3.1. Effect of CAP on Body Weight and Organ Index of Mice

As can be seen from Table 2, body weight was not significantly different between groups (*p* > 0.05). Compared to the CON group, liver indices increased significantly in the LPS group (*p* < 0.01). CAP tended to reduce this increase.

### 3.2. Effect of CAP on the Morphology of Mouse Liver and Intestinal Tissues

Figure 2 shows the histo-morphological observations of the mouse liver and colon. The liver lobules of the mice in the CON group had a clear structure, and the hepatic cords were arranged radially. Compared to the CON group, the LPS group showed structural disruption of the hepatic lobules, a reduced number and edema of hepatocytes, a dispersed arrangement of hepatocytes, and enlarged hepatic sinusoidal cavities. In contrast, the CAP group demonstrated a partial loss of the hepatic lobular structure and a small amount of hepatocyte edema. In the area of the colon, compared to the CON group, there was a decrease in goblet cells and an infiltration of inflammatory cells in the LPS group, and only some inflammatory cells were present in the CAP group. As shown in Appendix A, small intestinal microscopy showed that the duodenal villi of the CON group were broadly lobulated and well aligned, and most of the villi of the LPS group were ruptured with sparse crypts. Interestingly, CAP attenuated this damaging effect of LPS on the duodenum. In the jejunum and ileum, the villi and crypts of the CON group were neatly aligned, and CAP improved jejunal and ileal villi breakage caused by LPS. In addition, a significant increase in duodenal villus length and villous crypt ratio, as well as in jejunal villus length, was observed in the CAP group versus the LPS group (*p* < 0.01) (Appendix A). On the other hand, ileal crypt depth was significantly reduced in the CAP group (*p* < 0.01) (Appendix A).

### 3.3. Effect of CAP on Inflammatory and Oxidation/Reduction Markers in Mouse Liver and Colon

As illustrated in Figure 3, compared to the LPS group, IL-6, IL-1β, and TNF-α were significantly lower (*p* < 0.01) and IL-10 was significantly higher (*p* < 0.01) in the liver and colon of the CAP group.

To clarify CAP’s mitigating effect on mouse liver oxidative stress, liver oxidoreductase content was measured. According to Figure 4, in comparison with the LPS group, MDA levels were significantly lower in the CAP group (*p* < 0.05), and CAT, GSH-Px, and SOD activities were higher. However, the variations were not significant (*p* > 0.05).

### 3.4. Effect of CAP on the Expression of Anti-Inflammatory and Antioxidant Genes in Mouse Liver and Colon

As can be seen in Figure 5, compared with the LPS group, the expression of CAT in the liver was significantly increased in the group treated with CAP (*p* < 0.01). Liver GSH-Px and SOD were also increased. On the other hand, compared to the LPS group, IL-6 and IL-10 expression decreased and increased, respectively in the CAP group. Meanwhile, the expression of TNF-α and IL-1β was significantly reduced (*p* < 0.01).

In addition, colonic IL-10 expression was significantly higher in the CAP group than in the LPS and CON groups (*p* < 0.01). Meanwhile, the LPS group had a significantly increased colonic IL-6, TNF-α, and IL-1β expression, compared to the CAP group (*p* < 0.01).

### 3.5. Effect of CAP on the Alpha and Beta Diversity of Mouse Colonic Microbes

To assess the effect of CAP on the mouse colon microbiota, 16S rRNA high-throughput sequencing of mouse colon contents was performed, and community richness was characterized using Chao, Ace, and Sobs indices. All indices were significantly higher in the CAP (*p* < 0.01) and CON groups (*p* < 0.05) than in the LPS group (Appendix A–C). In addition, there were no significant differences in the Simpson and Shannon indexes among all groups (Appendix A). As shown in the Venn diagram, the three groups had 462 shared species and 275 endemic species at the OTU level, and the number of species in the CAP group was close to that in the CON group. In addition, the number of species shared by the CAP group with the CON and LPS groups (84 and 79, respectively) was more than twice that shared by the CON and LPS groups (30) (Appendix A).

Analysis of beta diversity revealed a partial aggregation of the CON group with the CAP and LPS groups, while the CAP and LPS groups were largely separated (Appendix A). Hierarchical cluster analysis showed that the CAP group was similarly clustered together with little variation (Appendix A).

### 3.6. Effect of CAP on the Microbial Composition of the Mouse Colon

Further analysis of the microflora of the mouse colon revealed that, at the level of the phylum, the relative abundances of the Desulfobacterota were higher (*p* < 0.05), while Deferribacterota and Campilobacterota were significantly higher in the CAP group than in the LPS group (*p* < 0.01). Likewise, compared to the CON group, both Deferribacterota and Patescibacteria were significantly increased (*p* < 0.05) in the CAP group, while Campilobacterota and Deferribacterota were significantly decreased (*p* < 0.05) in the LPS group (Figure 6A). 

A total of 185 genera were identified, including 32 differential genera among the groups (Figure 6B). The relative abundances of *Mucispirillum*, *Helicobacter*, *Prevotellaceae-UCG-001*, *Colidextribacter*, *unclassified-f-Oscillospiraceae*, *Muribaculum*, *norank-f-Desulfovibrionaceae*, *Eubacterium-brachy-group*, and *Odoribacter* were significantly increased in the CAP group than in the LPS group (*p* < 0.01 or *p <* 0.05). The relative abundances of *Monoglobus*, *Lachnoclostridium*, and *Eubacterium-brachy-group* were significantly higher (*p* < 0.05) in the CAP group than in the CON group. Additionally, *Prevotellaceae-UCG-001*, *Odoribacter*, *Colidextribacter*, and *unclassified f-Oscillospiraceae* were significantly more abundant in the CON group than in the LPS group (*p* < 0.05).

### 3.7. Effect of CAP on SCFAs in the Mouse Colon Tissue

The SCFA content in the mouse colon was examined. The results showed that the acetic, propionic, and butyric acid contents were significantly lower in both the CAP and LPS groups than in the CON group (Table 3). In addition, propionic acid levels were higher in CAP than in the LPS group; however, the difference did not reach statistical significance (*p* > 0.05).

### 3.8. Correlation Analysis of the Colon Microbiota with Inflammatory Factors, Oxidoreductases, and SCFAs

To further determine the correlation between altered colon microflora and oxidoreductase, inflammatory, and SCFA factors, the most abundant microbiota were correlated with hepatic oxidoreductase factors, hepatic and colon inflammatory factors, and colonic SCFA factors. As shown in Figure 7, liver MDA negatively correlated with *Desulfovibrio* and positively correlated with *A2*. *Odoribacter*, *Colidextribacter*, and *Candidatus-Arthromitus* were negatively correlated with hepatic and colon IL-1β, and positively correlated with IL-10. *Prevotellaceae-UCG-001* was negatively correlated with hepatic IL-1β, and *Oscillibacter* was negatively correlated with colon IL-1β. Additionally, *Odoribacter*, *Colidextribacter*, *Candidatus-Arthromitus*, and *Oscillibacter* were negatively correlated with colon IL-6, whereas *Colidextribacter* and *Candidatus-Arthromitus* were negatively correlated with hepatic IL-6. *Colidextribacter* and *unclassified-f-Oscillospiraceae* were negatively correlated with hepatic TNF-α, while *Odoribacter* and *Candidatus-Arthromitus* were negatively correlated with colon TNF-α. *Jeotgalicoccus* and *Staphylococcus* were positively correlated with colonic acetic, propionic, and butyric acids, and *Atopostipes* and *Sporosarcina* were positively correlated with butyric acid. *Lachnoclostridium* was negatively correlated with propionic and butyric acids, and *Eubacterium_ brachy_group* was negatively correlated with acetic acid and butyric acid.

## 4. Discussion

### 4.1. CAP Attenuates Hepatic Inflammation and Oxidative Stress in Mice by Reducing Liver Weight

CAP is a bioactive compound present in chili peppers and is widely used in various industries such as food, pharmaceuticals, healthcare, and biopesticides. Inflammation and oxidative stress may cause liver enlargement and weight gain, studies suggest [17,18], and CAP can reduce liver injury by inhibiting inflammatory responses and reducing oxidative stress [19]. In this study, LPS induction resulted in a significantly higher liver index in mice, and CAP tended to inhibit the LPS-induced increase in liver weight. Therefore, we hypothesized that the observed reduced liver indices in the CAP group may be related to the CAP-regulated hepatic inflammatory response and oxidative stress.

### 4.2. CAP Alleviates Oxidative Stress in Mice by Attenuating Hepatic and Intestinal Inflammatory Responses

Inflammation is an immune response that maintains tissue homeostasis in response to damage. However, if this irritation persists or does not subside over time, it can lead to disease [20]. Hepatic inflammation is a concerted response of various hepatocytes to hepatocyte death and inflammatory signals associated with intrahepatic injury or extrahepatic mediators from the gut–liver axis [21]. A previous study found a protective effect of CAP on LPS-induced liver tissue injury [14]. In the present study, CAP relieved hepatic inflammation by reducing hepatomegaly and by maintaining hepatic lobular morphology and structure. Differential changes in hepatic ILs are markers of the inflammatory response and important indicators of hepatic inflammation. CAP inhibited LPS-induced TNF-α and IL-1β in mice [22]. In the present study, CAP significantly decreased hepatic TNF-α, IL-1β, and IL-6 levels, and increased hepatic IL-10 levels. CAP also significantly increased the expression of CAT in the liver and significantly decreased the expression of TNF-α and IL-1β. These results suggest that CAP has an attenuating effect on the inflammation of the liver induced by LPS.

In the gastrointestinal tract, a persistent inflammatory response is a major cause of tissue damage, inflammatory bowel disease, and systemic inflammation [23,24]. Previous studies have shown that CAP inhibits intestinal inflammation [25]. In this study, CAP attenuated inflammatory cell infiltration in colonic tissues and ameliorated LPS-induced breakage of duodenal, jejunal, and ileal villi. CAP also increased the length of the duodenal and jejunal villi and decreased the depth of the ileal crypts, improving the LPS-induced histomorphology of the mouse small intestine.

Studies have also shown that CAP attenuates ulcerative colitis (UC) in mice by inhibiting the expression of oxidative stress proteins, proinflammatory cytokines, and TRPV1 [26]. In the present study, CAP significantly decreased the level of TNF-α, IL-1β, and IL-6 and increased the content and level of IL-10 in the colon. Overall, these results are an indication that CAP has an anti-inflammatory effect on the bowel.

Inflammation, altered redox homeostasis, and gut–liver axis dysfunction are major contributors to liver and intestinal disease [27]. At sites of inflammation, activated inflammatory cells induce tissue damage and oxidative stress by releasing enzymes, reactive substances, and chemical mediators [28]. This leads to increased levels of damage and ROS [29]. Oxidative stress can lead to differential expression of some genes involved in inflammatory pathways [30]. Therefore, a reduced inflammatory response may be associated with reduced oxidative stress. Studies have shown that CAP significantly ameliorates oxidative stress by enhancing GSH levels and increasing ROS and MDA [31]. In the present study, CAP significantly reduced hepatic MDA content and increased CAT, GSH-Px, and SOD enzyme activities. In addition, research has demonstrated a strong link between the enterohepatic axis and liver disease, and disruption of the enterohepatic axis contributes to liver inflammation and oxidative stress [32,33]. In addition, LPS-induced inflammation of the intestinal-hepatic axis has been well documented [34]. Therefore, attenuation of hepatic and intestinal inflammatory factors is important in alleviating oxidative stress. CAP could enhance antioxidant capacity by down-regulating the level of inflammatory factors, thereby maintaining the normal functioning of the organism.

### 4.3. CAP Alleviates Hepatic and Intestinal Inflammation and Oxidative Stress in Mice by Regulating the Colon Microbiota and SCFAs

The intestinal microbiota, which is altered by external factors, affects not only the intestinal tract but also various organs through a variety of pathways [35]. One study found that the intestinal microbiota mediates anti-inflammatory effects and reduces intestinal inflammation by modulating immunity and increasing the secretion of metabolic SCFAs [36]. The gut microbiota, which contains trillions of beneficial bacteria, plays a significant role in antioxidant production [37]. In this study, CAP significantly increased the abundance of *Mucispirillum* spp., which is associated with anti-inflammatory and antioxidant properties. Studies have shown that *Mucispirillum schaedleri*, the only representative genus of Deferribacteres, protects the host from colitis by interfering with pathogen invasion mechanisms [38,39]. In addition, Loy et al. [40] found that *M. schaedleri* can thrive under oxidative stress during inflammation by reducing nitrate and expressing specialized systems for scavenging oxygen and ROS in vivo. The Spearman’s correlation analysis showed that colonic *Odoribacter* spp. and *Colidextribacter* spp. were negatively correlated with inflammatory IL-6, IL-1β, and TNF-α, and positively correlated with IL-10. These findings are consistent with the results of Bingyong Mao and Changyu Wu [41,42], which suggests that CAP may reduce the level of inflammation and exert anti-inflammatory effects in the body by modulating the colonic microflora. With regard to exerting antioxidant effects, *Helicobacter* spp. can persist in the host and counteract oxidative stress through a number of different activities. For example, the antioxidant effect can be activated not only by increasing the activity of enzymes such as SOD and CAT but also by the family of peroxiredoxin proteins (alkyl hydroperoxide reductase, bacterial ferritin co-migrating proteins, and thiol peroxidases) function [43]. In this study, CAP significantly increased *Helicobacter* spp. abundance and hepatic CAT enzyme activities, suggesting that CAP may exert antioxidant effects by increasing *Helicobacter* spp. abundance. According to Fanghong Wang et al. [16], capsaicin could affect the composition and content of the intestinal flora in mice, and there were inconsistencies in the change trends of the intestinal flora between male and female mice. Therefore, the modulatory effect of CAP on the colonic microflora of mice in the present study may be related to their gender.

A key mechanism of metabolic regulation by the gut microbiota is through the production of short-chain fatty acids (SCFAs) [44]. SCFAs are fermentation products of intestinal microbiota, play an essential role in intestinal homeostasis, and are the main source of energy for colonocyte growth [45,46]. SCFAs also affect inflammation, oxidative stress, and microbial communities through complex regulatory mechanisms [47,48,49]. Xia Wen et al. [50] showed that propionate and butyrate exerted anti-inflammatory effects on Mycoplasma pneumoniae-stimulated THP-1 cells by inhibiting the expression of IL-4, IL-6, ROS, and NLRP3, and increasing the expression of IL-10 and IFN-γ. In addition, butyrate and propionate can regulate Keap1-Nrf2-dependent cell signaling pathways to maintain redox homeostasis through direct and indirect mechanisms [51,52,53]. In this study, CAP increased the amount of propionic acid. Studies have shown that SCFAs, especially propionate, possess antioxidant and anti-inflammatory effects [54], which suggests that CAP may alleviate body inflammation and oxidative stress by regulating propionic acid levels. Furthermore, in the correlation analysis of microflora with SCFAs, we found that *Jeotgalicoccus* spp. and *Staphylococcus* spp. were positively correlated with acetic and butyric acids. Longying Pei et al. [55] showed that the increase in SCFA content in the feces of mice that ingested *Morchella* spp. may be related to the production of *Jeotgalicoccus* spp. and other genera of bacteria. Similar to the results of previous studies, CAP reduced the levels of acetic and butyric acids in the present trial, which may be related to the fact that CAP reduced the abundance of harmful microflora in the colon. 

However, this paper is only a basic study of the effects of CAP on inflammatory and oxidative stress through intestinal microflora. Its specific mechanisms need to be further investigated in terms of the relevant signaling pathways et al. involved in the anti-inflammatory and antioxidant effects.

## 5. Conclusions

In summary, CAP reduced liver weight, enhanced the anti-inflammatory capacities of the liver and intestinal tract, alleviated hepatic oxidative stress, and regulated the colon microbiota in mice. However, how CAP exerts its anti-inflammatory and antioxidant effects by modulating the colonic microflora, mainly *Mucispirillum* spp. and *Helicobacter* spp. remains unclear, and future studies could continue to investigate its specific mechanism.

## Figures and Tables

**Figure 1 antioxidants-13-00942-f001:**
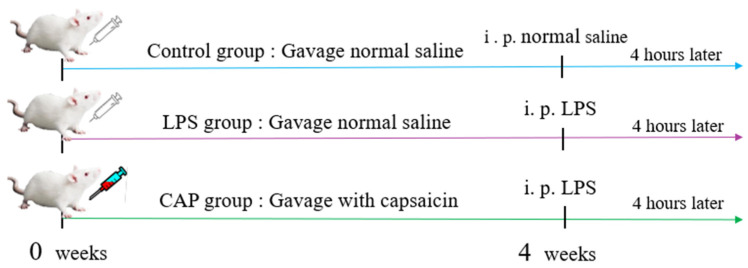
Experimental design. Equal doses of saline were administered daily to the CON and LPS groups, while CAP was administered to the CAP group. On the 28th day, LPS (10 mg/kg) was injected intraperitoneally in the LPS and CAP groups, and the CON group was injected with an equal volume of saline. Mice were then sacrificed after 4 h, and samples were collected.

**Figure 2 antioxidants-13-00942-f002:**
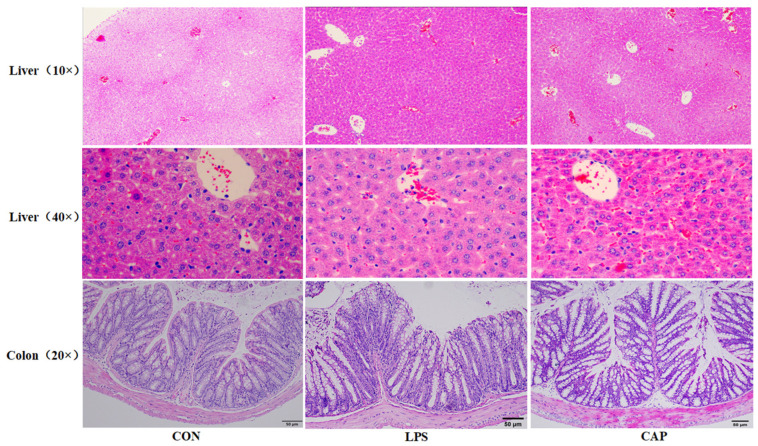
Effect of CAP on the morphology of mouse liver and colon tissues. The effect of CAP on the liver was assessed by H&E staining. Original magnification: ×100 and ×400; effect on the colon, original magnification: ×200.

**Figure 3 antioxidants-13-00942-f003:**
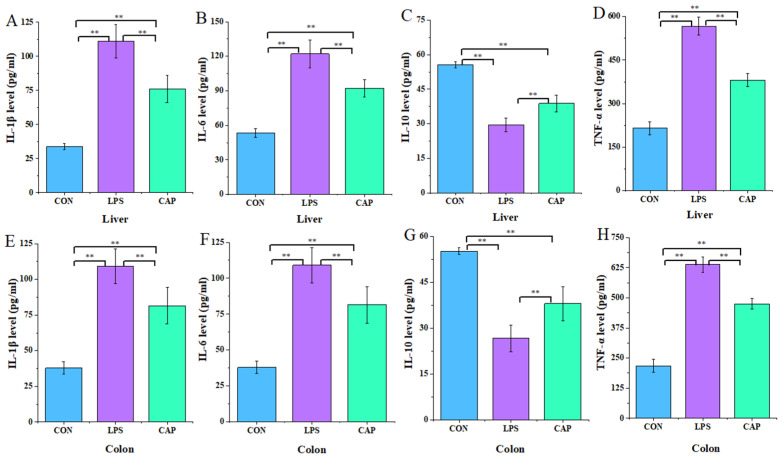
Effect of CAP on hepatic and colon inflammatory markers. Liver (**A**) IL-1β, (**B**) IL-6, (**C**) IL-10, and (**D**) TNF-α. Colon (**E**) IL-1β, (**F**) IL-6, (**G**) IL-10, and (**H**) TNF-α. Note: ** *p* < 0.01.

**Figure 4 antioxidants-13-00942-f004:**
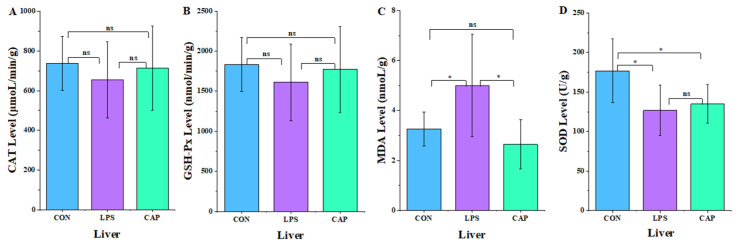
Effect of CAP on mouse liver oxidoreductase. (**A**) Catalase. (**B**) Glutathione peroxidase. (**C**) Malondialdehyde, (**D**) superoxide dismutase. Note: ns indicates that the difference was not significant (*p* > 0.05), * *p* < 0.05.

**Figure 5 antioxidants-13-00942-f005:**
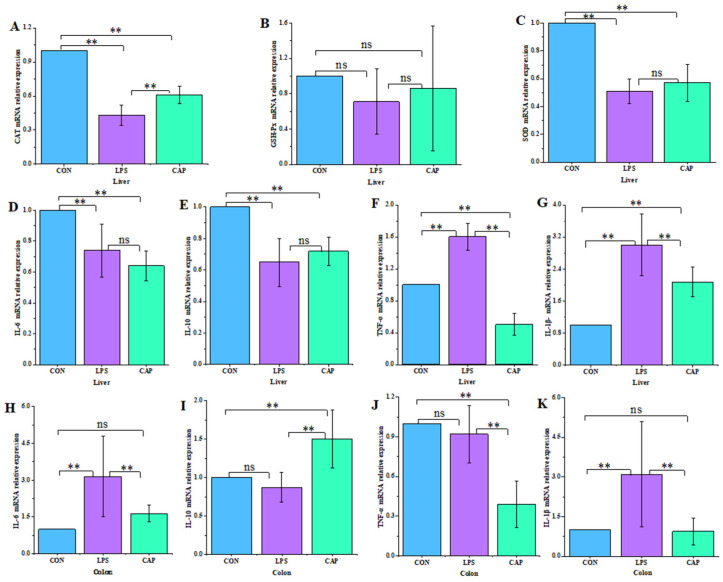
Effect of CAP on the expression of hepatic and colon antioxidant and anti-inflammatory markers. Liver (**A**) catalase, (**B**) glutathione peroxidase, (**C**) superoxide dismutase, (**D**) IL-6, (**E**) IL-10, (**F**) TNF-α, and (**G**) IL-1β. Colon (**H**) IL-6, (**I**) IL-10, (**J**) TNF-α, and (**K**) IL-1β. Note: ns indicates that the difference was not significant (*p* > 0.05), ** *p* < 0.01.

**Figure 6 antioxidants-13-00942-f006:**
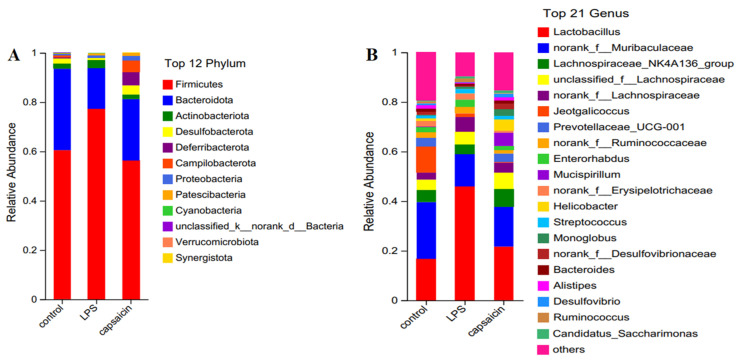
Effect of CAP on the microbial composition of mouse colon. (**A**) Phylum-level species composition. (**B**) Genus-level species composition.

**Figure 7 antioxidants-13-00942-f007:**
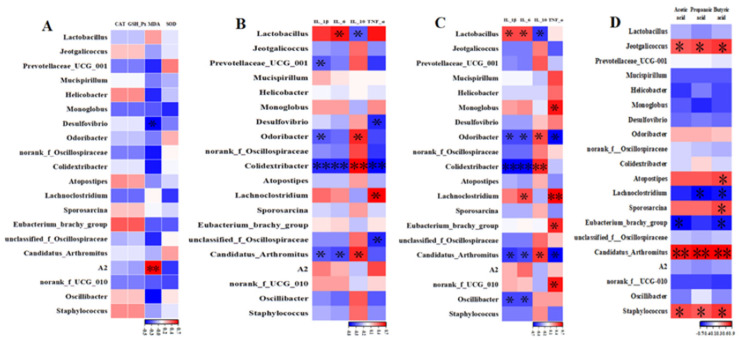
Correlation analysis of the colon microbiota with inflammatory factors, oxidoreductases, and SCFAs. (**A**) Correlation analysis between colon microbiota and hepatic oxidative reductivities sub. (**B**) Association between the colon microbiota and hepatic inflammatory factors. (**C**) Association between colon microbiota and colon inflammatory factors. (**D**) Association between colon microbiota and SCFA factors. Note: * Significant correlation at the 0.05 level (Double-tailed), ** Significant correlation at the 0.01 level (Double-tailed).

**Table 1 antioxidants-13-00942-t001:** Primer sequences.

Genes	Accession No.	Primer Sequences (5′-3′)	Size/bp
CAT	NM_009804.2	F: GGAGTCTTCGTCCCGAGTCT R: TGCCCTGGTCGGTCTTGTAAT	168
GSH-Px	NM_001329528.1	F: AGTGCGAAGTGAATGGTG R: TGTCGATGGTACGAAAGC	222
SOD	NM_011434.2	F: ATGGCGATGAAAGCGGTGTGC R: TTACTGCGCAATCCCAATCACT	465
β-actin	NM_007393.5	F: ATATCGCTGCGCTGGTCG R: GATCTTCTCCATGTCGTCCC	245
IL-1β	XM_006498795.5	F: TCCAGGATGAGGACATGAGCAC R: GAACGTCACACACCAGCAGGTTA	105
TNF-α	NM_001278601.1	F: TCTTCAAGGGACAAGGC R: GGACTCCGCAAAGTCTAA	251
IL-10	XM_036162094.1	F: GCTGGACAACATACTGCTAACC R: GAGGGTCTTCAGCTTCTCACC	178
IL-6	NM_001314054.1	F: TGATGGATGCTACCAAACTGGA R: TGTGACTCCAGCTTATCTCTTGG	197

**Table 2 antioxidants-13-00942-t002:** Effect of CAP on body weight and organ index in mice.

Group	CON	LPS	CAP	*p*-Value
Initial weight (g)	22.96 ± 1.18	22.94 ± 0.84	22.21 ± 115	0.411
Final weight (g)	33.57 ± 1.66	34.70 ± 1.61	34.50 ± 1.72	0.483
Liver index (%)	3.95 ± 0.27 ^Aa^	4.80 ± 0.35 ^Bb^	4.72 ± 0.13 ^Bb^	*p* < 0.001
Renal index (%)	1.15 ± 0.11	1.24 ± 0.28	1.32 ± 0.13	0.309
Spleen index (%)	0.33 ± 0.03	0.37 ± 0.05	0.41 ± 0.10	0.122
Thymus index (%)	0.26 ± 0.10	0.42 ± 0.16	0.29 ± 0.08	0.092

Note: Different capital letters in the shoulder scale of the same data indicate highly significant differences (*p* < 0.01). Different lower-case letters on the shoulders of the same data indicate significant differences (*p* < 0.05).

**Table 3 antioxidants-13-00942-t003:** Effect of CAP on LPS-induced colonic SCFAs in mice.

Item	Treatment		
	CON	LPS	CAP
acetic acid (µg/mL)	15.11 ± 3.05 ^Aa^	6.74 ± 0.25 ^Bb^	6.38 ± 2.84 ^Bb^
propionic acid (µg/mL)	4.47 ± 0.42 ^Aa^	3.30 ± 0.10 ^Bb^	3.39 ± 0.25 ^Bb^
butyric acid (µg/mL)	4.46 ± 0.60 ^Aa^	3.20 ± 0.05 ^Bb^	3.10 ± 0.25 ^Bb^

Note: Different capital letters in the shoulder scale of the same data indicate highly significant differences (*p* < 0.01). Different lower-case letters on the shoulders of the same data indicate significant differences (*p* < 0.05).

## Data Availability

The data presented in this study are available on request from the corresponding author.

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
