# Peer review of "Capsaicin Modulates Hepatic and Intestinal Inflammation and Oxidative Stress by Regulating the Colon Microbiota"

_antioxidants, 2024, doi:10.3390/antiox13080942_

Round 1

Reviewer 1 Report

The manuscript deals with timely and relevant subjects and methods. The authors have appropriately delineated the current state of the art.

The descriptions in the Materials and Methods section need additional edits. Moreover, some corrections are also appreciated in the section dealing with the description of the results. I suggest putting all the data and presentation of the results from Figure 7 in the Supplementary materials. In this way, it should be more impactful not only for the thorough review but also useful for all potential readers.

The discussion is clear and sufficiently provided. The conclusion, written with ambition, suggests that capsaicin could be potentially active in reducing inflammation and oxidative stress, based on the experiments performed and the results obtained. 

The manuscript has the potential to significantly contribute to the field of nutraceuticals. By bringing capsaicin to the forefront, this manuscript, once refined according to the suggestions, could be a significant step towards a better understanding of this biologically active compound derived from red hot chili peppers.

The sub-chapter of the Chemicals and Reagents (rows 67-68) is missing information about other consumables (reagents, chemicals) used. Additionally, catalogue and lot number should be listed for the capsaicin and LPS batches used in the experiment.

The sub-chapter 2.7. (row 101) - the title should be Liver and colon inflammatory biomarker assay; since intestinal is much broader term.

In general, the authors should put attention and carefully examine these terms (intestine and colon) in the whole manuscript and the precise term should be used when presenting the results which comprise intestine as such, or only colon.

The sub-chapter 2.11. (row 140) - it is not clearly stated that the total bacterial DNA was extracted from colon tissue; please specify what "colon content" means in this respect. 

Yet again, please put attention when presenting results for the colon or intestinal tissue. This is not completely clear in this section and needs more careful reading and correcting whenever needed. In example, in Figure 3 and Figure 4. The figures descriptions are mentioning "intestine", while in the drawings the "colon" is presented as a sample. 

Sub-chapter 3.8. (row269) - "effect of CAP on SCFAs in the Mouse Colon Tissue - the word "tissue" is missing, please add it.

Author Response

Please find the responses in the attachment

Reviewer 2 Report

I appreciate the work the study team invested in this research. Positive effects of Capsaicin on liver and intestinal lipopolysaccharide induced inflammation were demonstrated, as well its antioxidant properties. I read the iThenticate report in detail and it looks fine.

I have listed some comments for considerations below:

1.       Major:

a. The Authors should make it clearer that the positive effects of Capsaicin were made through regulating intestinal microbiota, as stated in the Title (Figure 6, especially from 6I onwards is not readable).

b. Animals and Experimental Design: Please mention mice gender. It is very important, as it may have important differences regarding the gender used (male or female or both).

c. Please include strength and limitations of your study.

d. Please include proper directions for future research (What should be done? How? etc).

a.       Please insert in Abstract brief info about mice groups, numbers, gender etc and interventions.

b.       Abstract: “CAP also decreased the overall content of short-chain fatty acids, except for propionic acid.” Reducing SCFAs is not a good effect, and this should be commented on in Discussion. (In Discussion, this aspect was approached, but it could be expanded more)

c.        Table 2 and Figure 2 – Legend: Please explain what is the role of “The following table is the same”.

d.       Figure 2 legend: If “the following table is the same” means the same interpretation of p values (ns indicates that the difference was not significant (p>0.05), * p < 0.05, ** p < 0.01), then this interpretation should be mentioned under every figure (Figure 3, Figure 4, Figure 5).

e.       Figure 2 – at 200% increase of the initial page, I can see the details, but not at the normal scale. Please correct.

f.         Figure 2 - Also, figure legend - why “n=6”?

g.        Table 3: Please insert interpretation under the table (as in Table 2).

h.       Discussion: Lines 311-315: “Studies have also shown that CAP attenuates ulcerative colitis (UC) in mice by inhibiting the expression of oxidative stress proteins….” – have no reference. Please add.

i.         Title: Maybe instead of “flora”, the current term “microbiota” could be used. Same for Abstract and sentences in the main text.

j.         References: Please review them attentively and make sure they are correctly reflecting the content in the main text (e.g. – neither [5], nor [6] – refer to UC, CD, colorectal cancer; moreover, [5] is related still to liver diseases, etc). Also, they should be updated.

Round 2

Reviewer 2 Report

I congratulate the authors for working hard in order to improve their manuscript. Now, it looks lear and tidy.

I have just three remarks:

11. Since only male mice were used, please insert the limitation of not using females, according to the recent literature (in Discussion).

22. Strengths and limitations were nicely written in the Response letter; however, they were not inserted in the manuscript. Please do so.

33. Same for the proper directions for future research.

Thank you

I have none. Thank you
